# Biomarker-Targeted Therapies in Non–Small Cell Lung Cancer: Current Status and Perspectives

**DOI:** 10.3390/cells11203200

**Published:** 2022-10-12

**Authors:** Haiyang Guo, Jun Zhang, Chao Qin, Hang Yan, Tao Liu, Haiyang Hu, Shengjie Tang, Shoujun Tang, Haining Zhou

**Affiliations:** 1Department of Thoracic Surgery, Suining Central Hospital, Suining 629099, China; 2Institute of Surgery, Graduate School, Chengdu University of TCM, Chengdu 610075, China; 3Institute of Surgery, Graduate School, Zunyi Medical University, Zunyi 563003, China

**Keywords:** non-small-cell lung cancer, targeted therapies, immunotherapies

## Abstract

Non-small-cell lung cancer (NSCLC) is one of the most common malignancies and the leading causes of cancer-related death worldwide. Despite many therapeutic advances in the past decade, NSCLC remains an incurable disease for the majority of patients. Molecular targeted therapies and immunotherapies have significantly improved the prognosis of NSCLC. However, the vast majority of advanced NSCLC develop resistance to current therapies and eventually progress. In this review, we discuss current and potential therapies for NSCLC, focusing on targeted therapies and immunotherapies. We highlight the future role of metabolic therapies and combination therapies in NSCLC.

## 1. Non-Small Cell Lung Cancer

In recent decades, lung cancer has decreased in incidence and mortality, but it remains the second most commonly detected cancer in the world [1]. Lung cancer can be divided into non-small cell lung cancer (NSCLC) and small cell lung cancer (SCLC) according to different pathological types, and NSCLC accounts for 80–85% of all lung cancers [2] (Figure 1).

The treatment of lung cancer involves surgery, radio-chemotherapy, immunotherapy, and targeted approaches with antiangiogenic monoclonal antibodies and tyrosine kinase inhibitors (TKIs) if tumors harbor a specific mutation [3]. NSCLC is often diagnosed at an advanced stage and has a poor prognosis [4]. Approximately 60% of patients with advanced NSCLC subtypes have specific molecular changes that may be suitable for targeted therapy [5].

Nowadays, next-generation sequencing technologies have enabled a comprehensive genetic characterization of NSCLC [6,7]. Therefore, clinicians have benefited from understanding cancer genetics.

At present, the actionable mutations with US Food and Drug Administration (FDA)-approved therapies include epidermal growth factor receptor (*EGFR*)-activating mutations, anaplastic lymphoma kinase (*ALK*) rearrangements, *ROS1* gene fusions, *BRAF^V600E^* mutations, and neurotrophic tropomyosin-related kinase (*NTRK*) gene fusions. Other emerging targeted therapies for NSCLC target gene aberrations associated with *ERBB2* (*HER2*), mesenchymal–epithelial transition (*MET*), rearranged during transfection (*RET*), *KRAS*^G12C^, NeuReGulin 1(*NRG1*), and the fibroblast growth factor receptor (*FGFR*) [8]. Other therapeutic targets include metabolic targets (*KEAP1-NFE2L2*, *STK11*) and immune targets (*PD-1*, *PD-L1*, *CTLA-4*).

## 2. Molecular Targets

### 2.1. EGFR

#### 2.1.1. *EGFR* p.L858R and *EGFR* Exon 19 Deletion

The *EGFR* gene is a tyrosine kinase receptor with a mutation rate of approximately 10% in patients with NSCLC (Figure 1) [11]. *EGFR* p.L858R and a 15 bp deletion in exon 19 are the two most frequent mutations in NSCLC, accounting for about 85–90% over all *EGFR* mutations [12].

The identification of *EGFR*-activating mutations in NSCLC drastically changed the disease’s clinical landscape. *EGFR* TKIs are now established first-line treatments for NSCLC with *EGFR* mutations. Compared to platinum-based chemotherapies, the primary *EGFR* TKIs showed advantages in progression-free survival (PFS) and overall survival (OS) (Table 1) [13,14]. Following the development of next-generation *EGFR*-targeted therapies, patients with *EGFR*-mutant tumors already have an OS of over three years [15].

However, the clinical efficacy of these initial inhibitors was ultimately limited by the development of drug resistance, in particular the T790M gatekeeper residue mutation [18]. In order to solve the problem caused by drug resistance, more irreversible mutant-selective inhibitors including afatinib [19], osimertinib [20], dacomitinib, and rociletinib [21] were developed as second or third *EGFR* inhibitors. Osimertinib, a third-generation TKI, was originally approved for the treatment of T790M-positive resistance [22]. In patients with first-generation *EGFR* TKI resistance caused by T790M, osimertinib demonstrated a superior objective response rate (ORR) and PFS than standard chemotherapy [22]. Therefore, it was recently approved for the first-line therapeutic drug of *EGFR*-mutant lung cancer [23].

Despite ORRs as high as 70–80% in the early stages of treatment, resistance inevitably occurs. Unfortunately, acquired resistance to osimertinib has been linked to mutations in *EGFR*, loss of T790M, and the bypass of pathways via *KRAS*, *MET*, and *PIK3CA* [24,25]. Targeted therapeutic strategies are being evaluated in numerous trials following *EGFR* TKI resistance. In the current environment, cytotoxic chemotherapy is the most common treatment option. Notably, immune checkpoint inhibitor therapy is lacking in efficacy in *EGFR*-mutant NSCLC [26]. There are a large number of third-generation *EGFR* TKIs which have been developed, such as nazartinib (NCT02335944), lazertinib (NCT04248829, NCT04077463, NCT04487080) and aumolertinib (NCT04923906), in ongoing clinical trials [27].

#### 2.1.2. EGFR Exon 20 Insertion Mutations

Exon 20 of the *EGFR* kinase domain contains around 10% of the *EGFR* activating mutations (*EGFR* ins20) [28] (Figure 1). Early generations of *EGFR* TKIs did not show the same sensitivity to *EGFR* ins20 mutations as classical activating *EGFR* mutations.

With significant advancements in sequencing technology, the success gained in overcoming these systemic hurdles has substantially increased the speed of recent breakthroughs in *EGFR* ins20 targeted medicines. An example of this is the recent approval of amivantamab by FDA [29] (Figure 2).

A recent retrospective study of 6290 NSCLC patients found that patients with *EGFR* ins20 tumors had longer OS and time to treatment cessation with platinum chemotherapy compared with patients without targeted mutations [30].

#### 2.1.3. Other EGFR Mutations

A number of unusual *EGFR* mutations have also been identified, including G719X, S768I, and L861Q. A multicenter, phase II study, in which 37 patients were enrolled, demonstrated favorable activity with manageable toxicity of osimertinib used for those with harboring uncommon mutations. These patients were given osimertinib as their first-line therapeutic drug in 61% of cases. L861Q, S768I, and other mutations were the next most common, followed by G719X. The ORR was 50% (95% CI, 33% to 67%). The median PFS was 8.2 months (95% CI, 5.9 to 10.5 months) [31].

A recent extensive study discovered a novel atypical *EGFR* mutation named very rare *EGFR* mutations. In this study, researchers found that treatment responses in very rare *EGFR* mutations were highly heterogeneous. It is worth noting that there was no significant tendency for co-occurring TP53 mutations to adversely affect the outcome in *EGFR*-TKI-treated patients in the exon 20 insertion mutations group or the in very rare *EGFR* mutations group [32].

### 2.2. ALK

The first case of NSCLC with *ALK* rearrangement was reported in 2007 [33]. *ALK* is a tyrosine kinase that belongs to the insulin receptor superfamily of transmembrane receptors [34]. *ALK*- rearrangement NSCLC accounts for approximately 3–8% of NSCLC [35] (Figure 1).

Similar to *EGFR*-targeted therapy, TKI-targeting *ALK* is the first-line therapeutic drug for *ALK*-rearranged advanced NSCLC. Since the first *ALK*-TKI crizotinib was developed, patient prognosis has improved considerably [36]. Currently, five TKIs are approved by the FDA. Since the FDA approval of crizotinib [37] in 2011, ceritinib [38] was approved in 2014, alectinib [39] in 2015, brigatinib [40] in 2017, and lorlatinib [41] in 2018 (Figure 2).

Of these, crizotinib is the most widely investigated. Several clinical trials including at least two-phase III trials showed that crizotinib was superior to chemotherapy in patients with advanced NSCLC that were positive for *ALK* rearrangements [42,43] (Table 2).

Subsequently, PFS benefit was reported in phase III trials for several next-generation *ALK* TKIs in treatment-naive patients. Ceritinib and chemotherapy were compared [44]. Subsequently, other *ALK* TKIs, alectinib [45], brigatinib [46], ensartinib [48], and lorlatinib [47] were compared with crizotinib. Following the experience with osimertinib in NSCLC patients with *EGFR* mutations, the second-generation *ALK* TKIs have replaced crizotinib as the first-line treatment for advanced-stage *ALK*-rearranged NSCLC.

In ASCEND-4 [44], of 376 untreated patients, 189 received ceritinib and 187 received chemotherapy. Ceritinib significantly enhanced PFS compared with platinum-based chemotherapy (median PFS, 16.6 months vs. 8.1 months, respectively; HR, 0.55 [95% CI, 0.42–0.73]; *p* < 0.00001).

Researchers randomly assigned patients with metastatic lung cancer who expressed the *ALK* gene to receive either alectinib or crizotinib as a first therapeutic drug in a phase III randomized study (*n* = 303). Compared with crizotinib, alectinib markedly improved the rate of investigator-assessed PFS (12-month event-free survival rate, 68.4% vs. 48.7%, respectively; HR, 0.47 [95% CI, 0.34–0.65]; *p* < 0.001) [45].

Based on these data, in the main therapy of *ALK*-positive NSCLC, alectinib exhibited greater effectiveness and decreased toxicity.

The randomization of 275 patients in another phase III trial found that 137 received brigatinib and 138 received crizotinib. The median follow-up in the brigatinib group was 11.0 months and 9.3 months in the crizotinib group in the first interim analysis (99 incidents). Compared with crizotinib, brigatinib had a higher rate of PFS (estimated 12-month PFS of 67% vs. 43%; HR of 0.49 [95% CI, 0.33–0.74]; *p* < 0.001) [46].

Based on these data, brigatinib outperforms crizotinib in terms of effectiveness, tolerability, and quality of life, making it a viable first-line therapeutic drug for *ALK*-positive NSCLC.

Despite the fact that alectinib and brigatinib have a median duration of chronic control of more than two years, drug resistance persists.

Based on previous data, initial phase III results with lorlatinib, an *ALK*-TKI of the third generation, showed promising results in this setting. In this trial, patients were randomly assigned to receive first-line lorlatinib (*n* = 149) or crizotinib (*n* = 147). The ORR was 76% vs. 58%, and 12-month PFS was 78% vs. 39%; HR, 0.28 [95% CI, 0.19–0.41]; *p* < 0.001 [47]. These results may redefine the new potential standard of care in the first-line setting.

In an open-label, multicenter phase III study, ensartinib, an oral tyrosine kinase inhibitor of *ALK*, showed systemic and central nervous system improvement for patients with *ALK*-positive NSCLC. In this trial, 290 patients were randomized to accept ensartinib or crizotinib. In the intent-to-treat population, ensartinib had a considerably longer median PFS than crizotinib (25.8 vs. 12.7 months; HR, 0.51 [95% Cl, 0.35–0.72]; *p* < 0.001) [48]. Ensartinib did not achieve its median PFS in the modified intention-to-treat population, whereas crizotinib group was 12.7 months (95% CI, 8.9–16.6 months; HR, 0.45 [95% CI, 0.30–0.66]; *p* < 0.001). Ensartinib did not improve PFS for patients without brain metastases compared to crizotinib, which achieved 16.6 months (at 12 months: 4.2% with ensartinib vs. 23.9% with crizotinib; cause-specific HR, 0.32 [95% CI, 0.16–0.63]; *p* = 0.001) [48].

Based on these data, a new treatment option available for patients with NSCLC who were *ALK*-positive was ensartinib.

Approximately 50–60% of patients with *ALK* kinase-domain mutations are resistant to second-generation *ALK* TKIs [49]. It may be important to note that *ALK* fusion variants may play a role in the development of *ALK* inhibitor resistance on-target [50]. Several recurrent acquired mutations in *ALK* confer resistance by decreasing drug binding, including I1171T/N/S, V1180L, G1269A and G1202R [51]. Studies have shown that *ALK*^G1202R^ mutations are more common in tumors harboring EML4-*ALK* variant 3 (exon 6a/b of EML4 fused with exon 20 of *ALK*) than variant 1 (exon 13 of EML4 fused with exon 20 of *ALK*) [52].

There is no evidence that other TKIs other than lorlatinib can overcome resistance associated with G1202R [53]. Consistent with its broad *ALK* mutation coverage and CNS penetration, loratinib showed significant overall and intracranial activity *ALK* tyrosine kinase inhibitor in both *ALK*-positive NSCLC primary patients and in crizotinib second-generation therapy [41]. Moreover, loratinib has a robust and durable response and a high objective intracranial response in previously treated Chinese patients with *ALK*-positive NSCLC [54]. However, approximately 25–30% of patients who are treated with alectinib develop G1202R resistance, while 10–15% have I1171X resistance [49].

The latest *ALK* inhibitor developed by researchers is XMU-MP-5, which is designed to overcome crizotinib resistance mutations, such as L1196M and G1202R. It has achieved promising results in in vitro experiments [55].

Neuroendocrine transformation (*NET*) is another mechanism of *ALK* TKI resistance in NSCLC. In a study of 15 *NET* patients, immunohistochemistry showed p53/Rb inactivation in 72.7% of pre-*NET* lesions. Therefore, for patients with *NET* progression, p53/Rb risk prediction should be performed in addition to genomic assessment [56].

Up to now, in first-line therapy, there are already many *ALK* TKI alternatives, but the best upfront treatment option remains controversial [57,58].

A patient with metastatic *ALK*-rearranged NSCLC acquired crizotinib resistance due to a mutation in the *ALK* kinase domain, as shown in an investigation. This acquired mutation conferred lorlatinib resistance, but it also restored sensitivity to crizotinib [59].

There are no firm findings on the treatment sequence after first-line therapy, and further research is needed.

### 2.3. ROS1

Similar to *ALK*, *ROS1* possesses tyrosine kinase activity and is a member of the insulin receptor family [60]. NSCLC was the second solid tumor found to harbor *ROS1* rearrangements. Rearrangement of this gene is found in 1–2% of NSCLC, but more frequently in young women with adenocarcinoma who never smoked [61] (Figure 1).

The *ROS1* gene was first implicated in chromosomal rearrangements in NSCLC in 2007 [62]. Since then, oncogenic *ROS1* rearrangements have become a well-established therapeutic target in NSCLC.

The FDA-approved crizotinib as the first TKI to treat *ROS1*-rearranged NSCLC (Figure 2). This approval was based on efficacy and safety results from the expansion cohort of the phase I crizotinib study, which revealed an ORR of 72% and a median PFS of 19.2 months in advanced *ROS1*-rearranged NSCLC [61].

East Asian patients with *ROS1*-positive advanced NSCLC who had undergone three or fewer lines of previous systemic therapy were included in a phase II, open-label, controlled trial. A total of 127 individuals were included in the evaluations of effectiveness and safety, with 49.6% still taking medication at the end of the data collection period. With 17 full replies and 74 partial responses, the ORR was 71.7% (95% CI, 63.0–79.3%) [63]. Researchers reported similar ORR of 70% (95% CI, 51–85%) and median PFS of 20.0 months (95% CI, 10.1-not reached) for 34 European patients with advanced *ROS1*-positive NSCLC in a recent single-arm, multicenter, phase II trial [64].

In spite of the high response rates of crizotinib, the majority of patients eventually succumb to disease progression in part as a result of inadequate drug CNS penetration and/or the development of *ROS1* resistant mutations [65].

The second-generation TKI ceritinib is indicated for the treatment of NSCLC that is resistant to crizotinib and those that have not been treated with TKIs. There was a 67% ORR (95% CI, 48–81%) with a 21-month DOR (95% CI, 17–25 months) and a 19.3-month median PFS (95% CI, 1–37 months) in a phase II trial of 30 crizotinib-naive patients with *ROS1*-positive NSCLC [66]. Despite the fact that ceritinib may be a promising treatment option for *ROS1*-positive NSCLC based on this study, several crizotinib-resistant *ROS1* mutations such as G2032R, D2033N, L1951R, and S1986Y/F cannot be overcome by ceritinib [67,68].

*ROS1* and *ALK* TKI lorlatinib displays strong penetration into the CNS and is effective against a wide range of *ROS1* resistance mutations in preclinical studies.

Currently, during the first-in-man phase I multicenter study, 12 patients with *ROS1*-positive NSCLC received at least one dose of lorlatinib. Finally, 6 (50%) of the 12 patients obtained an objective response (95% CI, 21–79). Meanwhile, lorlatinib showed both systemic and intracranial activity in patients with advanced *ROS1*-positive NSCLC [69]. In one of the latest studies, when compared to the crizotinib group, the ORR of lorlatinib was higher (76% [95% CI, 68–83%] vs. 58% [95% CI, 49–66%]) [47]. Among those with measurable brain metastases, the intracranial response was 82% (95% CI, 57–96%) of loratinib group and 23% (95% CI, 5–54%) of the crizotinib group [49]. In particular, it was found that 71% of patients treated with lorlatinib had an intracranial complete response [49]. However, since lipid levels are often altered with lorlatinib, adverse events were more common than with crizotinib [47].

Entrectinib is a multikinase inhibitor with activity against *ROS1*, *ALK*, and *TRK* [70]. A substantial concentration of it reaches the central nervous system because of its ability to cross the blood-brain barrier [71]. An integrated review of three phase I-II trials showed that entrectinib is active and provides long-term disease control in patients with *ROS1* fusion-positive NSCLC. It is well tolerated and has a controllable safety profile, making it suitable for long-term therapy in these patients [72].

### 2.4. BRAF

The *RAS-RAF-MEK-ERK* signaling pathway is one of the most frequently mutated in human tumors [73]. *BRAF* is one of the most frequently mutated kinases in human cancer, especially melanoma, with activating mutations in *BRAF* seen in 40–50% of malignancies. Smokers are more likely to have somatically activating *BRAF^V600E^* mutations, which have been reported in 1–2% of NSCLC [35] (Figure 1).

*BRAF* mutations do not coincide with other oncogenic alterations in a significant number of lung adenocarcinomas.

As the use of next-generation sequencing in clinical practice becomes more frequent, oncologists often find *BRAF^non-V600^* mutations in their patients’ malignancies, but they are unsure of feasible therapeutic alternatives for optimum therapy. Dabrafenib or vemurafenib, which are single-agent *BRAF* inhibitors, have had rather short-term responses [74,75]. In patients with metastatic *BRAF^V600E^* mutant NSCLC, 59 patients were enrolled in a phase II study investigating combination pathway blocking with dabrafenib (a *BRAF* inhibitor) and trametinib (a *MEK* inhibitor). This study was successful, with a response rate of 64% and a median PFS of 11 months [76].

These data backed up a molecularly focused approach for patients with *BRAF*-mutant lung cancer. The FDA has approved dabrafenib and trametinib for the treatment of metastatic NSCLC exhibited the *BRAF^V600E^* mutation [77] (Figure 2).

Another trial launched by the French National Cancer Institute (INCA) assessed the effectiveness and safety of vemurafenib in tumors with diverse BRAF mutations [78]. The trial recruited 118 patients (101 with *BRAF^V600E^* mutations and 17 with *BRAF^nonV600E^* mutations), with a median follow-up of 23.9 months. There was no objective response in the *BRAF^nonV600E^* cohort, hence it was halted. 43/96 individuals in the *BRAF^V600E^* cohort exhibited objective responses. The median response duration was 6.4 months, while the median PFS was 5.2 months (95% CI, 3.8–6.8 months), and the median OS was 10 months (95% CI, 6.8–15.7 months). The safety profile of vemurafenib was consistent with earlier studies [78]. These data indicated that *BRAF^V600E^* mutations should be a routine biomarker screened of NSCLC, and vemurafenib may be the novel *BRAF*-mutation targeted drug.

### 2.5. MET

In epithelial cells, the *MET* receptor plays a role in tyrosine kinase signaling [79].

*MET* exon 14 skipping mutations are seen in 3–4% of patients with adenocarcinomas and 1–2% of individuals with another NSCLC histology [80] (Figure 1).

*MET* TKIs are classified as type I, II and III. Inhibitors of type Ia (such as crizotinib) block ATP binding to prevent receptor phosphorylation/activation. Capmatinib, tepo-tinib, savolitinib, and AMG 337 are type Ib inhibitors that are more effective against *MET* than type Ia inhibitors. Unlike ATP-binding sites, type III inhibitors (such as tivantinib) bind to allosteric sites [81].

Capmatinib was tested in 364 patients with stage IIIB/IV NSCLC in the GEOMETRY mono-1 trial. Patients were divided into groups depending on their past treatment regimens and *MET* status. In patients with a *MET* exon 14 skipping mutation, the study found that overall, 41% (95% CI, 29–53%) of 69 patients who had previously had one or two lines of medication responded, compared to 68% (95% CI, 48–84%) of 28 patients who had never received treatment. Furthermore, the median duration of response (DOR) was 9.7 months [82].

Based on these data, the FDA has approved capmatinib for treating patients with metastatic NSCLC with an exon 14 skipping mutation (Figure 2).

Tepotinib monotherapy was investigated prospectively in individuals with advanced NSCLC with *MET* exon 14 skipping mutations in the VISION open-label Phase II trial. At least 9 months of follow-up were conducted on 152 patients who received tepotinib. According to an independent analysis, based on the combined biopsy group, 46% of patients responded (95% CI, 36–57%) and the median DOR was 11.1 months (95% CI, 7.2 to NE).

The response rate for 66 patients in the liquid biopsy group was 48% (95% CI, 36–61%) and for 60 patients in the tissue biopsy group was 50% (95% CI, 37–63%). Both approaches had favorable outcomes in 27 cases. Regardless of previous therapy for advanced or metastatic cancer, the investigator-assessed response rate of 56% (95% CI, 45–66%) was identical [83].

Based on these data, tepotinib was granted breakthrough therapy designation by the FDA in September 2019 for the treatment of metastatic NSCLC patients with *MET* exon 14 skipping mutations who progressed following platinum-based cancer therapy (Figure 2).

Crizotinib was used in the METROS single-arm research, which included 26 patients (each) in two cohorts having *ROS1* or *MET* rearrangements and one or more previous chemotherapy regimens. FISH was used to centrally evaluate *ROS1* rearrangement and *MET* amplification. The researchers used direct sequencing to confirm the *MET* mutation at the end of the trial. The major trial endpoint was RR, which was 27% (of those with *MET*), with a PFS of 4.4 months and 13 patients reporting SAEs (50%) [69].

Treatments for patients with *MET* exon 14 skipping mutations are being revolutionized by *MET* TKIs. However, resistance to *MET* TKIs limits the magnitude and DOR. Several mechanisms of resistance to *MET* TKI have been established, such as targeted resistance driven by mutations in kinase structural domains that affect drug-receptor binding or its ATP affinity, amplification of *MET* exon 14 mutant alleles, and activation-mediated off-target resistance bypass signaling [84,85,86,87]. Therefore, to effectively target NSCLC, it is crucial to identify the mechanisms of resistance to *MET* TKIs.

### 2.6. RET

In 1–2% of NSCLC patients, *RET* gene rearrangements can enhance tumor development [88] (Figure 1). Patients with nondriver mutation NSCLCs, such as those with ROS, are younger and more likely to be light smokers or non-smokers.

Multiple TKIs that target *RET* have shown minimal therapeutic benefit in patients with *RET* rearrangements in their malignancies, includes cabozatinib [89] (tumor response rate of 28% and median PFS of 5.5 months) and vendatinib [90] (response rate of 18% and median PFS of 4.5 months).

For patients with advanced *RET* fusion and previous platinum-based therapy, a phase II study showed that selpercatinib had a long-term effectiveness. The ORR was 64% (95% CI, 54–73%) in the first 105 patients with *RET* fusion-positive NSCLC who had previously received at least platinum-based chemotherapy. The median response time was 17.5 months (95% CI, 12.0 could not be evaluated), and 63% of the responses were still active after 12.1 months of follow-up. The number of patients who had an objective response was 85% (95% CI, 70–94%) among 39 previously untreated patients, and 90% of the responses continued for 6 months. The percentage with an objective intracranial response was 91% (95% CI, 59–100%) among 11 patients with demonstrable central nervous system metastases at enrolment. At the same time, the toxic effects of selpercatinib were low-grade [91].

Based on these data, selpercatinib had durable efficacy for patients with *RET* fusion-positive NSCLC.

Investigators presented results from ARROW, a continuing noncomparative phase I or II study with pralsetinib, at the ASCO 2020 (and 2019) conference. ARROW is a phase 1/2 multi-cohort, open-label trial conducted at 71 locations (community and academic cancer centers) in 13 countries. From 17 March 2017 through 22 May 2020, pralsetinetib was given to 92 patients with *RET* fusion-positive NSCLC, and 29 patients who had never received treatment were given the drug by 11 July 2019. OS was documented in 53 (61%; [95% CI, 50–71]) of 87 patients who had previously received platinum-based chemotherapy, with 5 patients achieving a full response, and 19 (70%; 50–86) of 27 treatment-naive patients achieving a complete response [92].

Based on these data, FDA approval to pralsetinib for adult patients with metastatic *RET* fusion-positive NSCLC (Figure 2).

### 2.7. NTRK1

The *NTRK1* gene encodes tropomyosin-related kinase A [93]. *NTRK* gene abnormalities have been found in fewer than 1% of NSCLC [94] (Figure 1).

The *NTRK* mutation treatment drug larotrectinib was developed using a clinical trial that enrolled patients with any kind of *NTRK* gene rearrangements which showed a tumor response rate of 75% and 12-month PFS rate of 55%. As a result, larotrectinib was approved for treatment of *NTRK*-altered cancers regardless of their primary site by FDA (Figure 2). FDA approved the case series in which four patients with lung cancer were included as part of the FDA approval process [95].

Entrectinib was administered to patients with *NTRK*-positive tumors of various histology in a second pooled analysis, and Doebele and colleagues reported that 31 (57%; [95% CI, 43.2–70.8%]) of 54 patients treated with entrectinib achieved an objective response, with a median PFS of 11.2 months (95% CI, 8.0–14.9 months) and a median DOR of 10.4 months (95% CI, 7.1 to not estimable) [96].

The FDA approved entrectinib in August 2019 for the therapeutic drugs of metastatic *ROS1*-positive NSCLC and *NTRK*-positive tumors in both adults and children (Figure 2). Since larotrectinib was approved by the FDA in November 2018 (Figure 2), two TKIs have been licensed for NTRK gene fusion-positive solid tumors.

Most patients take first-generation *TRK* inhibitors well, with toxicity profiles typified by infrequent off-tumor, on-target side effects (attributable to *TRK* inhibition in non-malignant tissues). Despite the fact that many patients have long-term disease control, advanced-stage *NTRK* fusion-positive malignancies eventually develop resistance to *TRK* inhibition. Resistance might be caused by the acquisition of *NTRK* kinase domain mutations.

Following that, second-generation *TRK* inhibitors such as LOXO-195 and TPX-0005, which are now in clinical studies, can overcome some resistance mutations [97,98].

### 2.8. KRAS

The *KRAS*^G12C^ mutation is present in approximately 13% of patients with NSCLC [99] (Figure 1).

Because of the lack of a deep binding pocket, NSCLC with oncogenic *KRAS* mutations is notoriously hard to treat with small molecule inhibitors [100]. Sotorisib, a first-in-class *KRAS*^G12C^ small molecule inhibitor, has been approved for irreversible inhibition of *KRAS*^G12C^ via an interaction with the P2 pocket by the FDA. Sotorasib showed a 37% ORR and a median PFS of 6.7 months in the Code Brea K 100 phase II single-arm study [101]. KRYSTAL-1 phase I and II data, which have demonstrated a 45% ORR, have led the FDA to designate adagrasib as a breakthrough therapeutic drug [102] (Figure 2).

These data showed that combinatorial techniques can help increase response persistence, and several studies are currently underway to investigate the impact of checkpoint inhibitors and other *MAPK* pathway inhibitors.

### 2.9. HER2

*HER2* mutation have been found in approximately 2% of NSCLC [103] (Figure 1).

Trastuzumab emtansine(T-DM1), a HER2-targeted antibody-drug combination, has demonstrated tumor responses in these patients (*n* = 18; response rate of 44% and median PFS of 5 months) [104].

Meanwhile, T-DM1 was studied in a phase II study in patients with advanced *HER2*-overexpressing NSCLC who had previously been treated. In this study, 59 patients received T-DM1 (29 IHC 2+, 20 IHC 3+). In the IHC 2+ cohort, no treatment responses were detected. In the IHC 3+ cohort, four partial responses were observed (ORR, 20% [95% CI, 5.7–43.7%]). The clinical benefit rates in the IHC 2+ and 3+ groups were 7% and 30%, respectively. Respondents took 2.9, 7.3, 8.3, and 10.8 months to complete their responses. Between groups, median PFS and OS were similar [105].

Seventy-eight individuals were enrolled in a multicenter study to assess effectiveness and safety. In this study, the 6-month PFS rate was 49.5% (95% CI, 39.2–60.8%). Pyrotinib had 19.2% of ORR (95% CI, 11.2–30.0%), median OS of 10.5 months (95% CI, 8.7–12.3 months), and median PFS of 5.6 months (95% CI, 2.8–8.4 months). Meanwhile, there was a median DOR of 9.9 months (95% CI, 6.2–13.6 months). With *HER2* mutations in exon 20 and without exon 20, ORRs were 17.7% and 25.0%, respectively. Afatinib exposure and brain metastases at baseline had no effect on ORR, PFS, or OS. As the illness progressed, the loss of *HER2* mutations and the development of amplification in *EGFR* and *HER2* were discovered [106].

Based on these data, in NSCLC with exon 20 or non-exon 20 *HER2* mutations, pyrotinib showed promising effectiveness and tolerable tolerability.

Trastuzumab (T-DXd) is a novel antibody–drug conjugate that targets *HER2*. In an ongoing multicenter study called DESTINY-Lung01, the effectiveness of T-DXd in treating patients with non-squamous NSCLC overexpressing *HER2* or containing a *HER2*-activating mutation is being tested [107]. The ORR was 61.9% (95% CI, 45.6–76.4%) at data cutoff. The following results confirm its long-lasting anti-cancer activity: the median DOR was 9.3 months (95% CI, 5.7–14.7 months), median PFS was 8.2 months (95% CI, 6.0–11.9 months) and median OS was 17.8 months (95% CI, 13.8–22.1 months) [108]. These ideal outcomes demonstrated durable responses in NSCLC with *HER2*-mutation.

Recently, FDA granted accelerated approval to fam-trastuzumab deruxtecan-nxki for *HER2*-mutant NSCLC (Figure 2). Efficacy for accelerated approval was based on DESTINY-Lung02 (NCT04644237). The available, substantial results showed that the confirmed ORR was 58% (95% CI, 43–71 months) and the median DOR was 8.7 months (95% CI, 7.1 months to NE).

## 3. Molecular Targets with Potential Therapeutics

### 3.1. NRG1

Studies have shown that *NRG1* fusions are another rare but potentially actionable class of alterations in NSCLC. However, *NRG1* fusions are rare overall (<1% in NSCLC) [109] (Figure 1).

Given the rarity of *NRG1* fusions, clinical experience with *NRG1*-directed therapies remains limited. Although anti-*ERBB3* mAb therapy (GSK2849330) achieved a durable response in abnormal responders to NRG1 rearranged IMA in an in vitro assay, a trial indicated that no response to anti-*ERBB2* treatment (afatinib) was achieved in four *NRG1* rearranged IMA patients (including the index patient after GCK2849330) [110]. Tarloxotinib is a prodrug that takes advantage of tumor hypoxia to produce high levels of the potent covalent pan-HER tyrosine kinase inhibitor tarloxotinib effector (tarloxotinib-E) in the tumor microenvironment. A recent trial tested the clinical response to tarloxotinib [111].

### 3.2. FGFR

The fibroblast growth factor receptors (*FGFR*s) are involved in cell growth, development, and cancer growth. NSCLC has an amplification of *FGFR*1 in about 0.2% of cases, which raises the possibility that the novel anti-cancer drug could target *FGFR* [112] (Figure 1). However, acquired resistance to this type of treatment remains a significant therapeutic issue.

To establish whether *FGFR* is a potential target, researchers examined at the response of NSCLC cell lines to the new selective *FGFR* inhibitor CPL304110 and the probable mechanism of acquired resistance. In this study, despite considerable genetic variations between CPL304110-sensitive and resistant cell lines, as well as elevated p38 expression/phosphorylation, they also observed increased *MAPK* gene expression. They discovered that inhibiting p38 restored CPL304110 sensitivity in these cells. Furthermore, the p38 *MAPK*-overexpressing parental cells showed reduced sensitivity to inhibition of *FGFR*, suggesting that p38 *MAPK* is responsible for resistance to a new *FGFR* inhibitor [112].

Based on these data, the p38 *MAPK* pathway may point to a new avenue for NSCLC targeted treatment.

## 4. Metabolic Targets

### 4.1. KEAP1-NFE2L2

Activation of the *KEAP1-NFE2L2* pathway increases tumor growth and aggressiveness [113,114]. *KEAP1-NFE2L2* mutations are estimated to be found in 23% of NSCLC based on the Cancer Genome Atlas Network (TCGA) [9] (Figure 1).

In recent years, *KEAP1-NFE2L2* mutations have been found to have multiple prognostic implications. *KEAP1-NFE2L2* was shown to be a predictive biomarker of radiation resistance [115]. In addition, *KEAP1-NFE2L2* deletion has been identified as a mechanism of acquired resistance to targeted therapy in *EGFR*-mutant tumors [116].

As the prognostic impact mutations of *KEAP1-NFE2L2* continue to evolve, many experiments have sprung up to develop therapies targeting this pathway. The compound sapanisertib (TAK-228), which inhibits both mTORC1 and mTORC2, was the first therapy for cancer patients who carried *KEAP1-NFE2L2* mutations [117]. A study of cancer models performed in squamous cell lung showed that *KEAP1-NFE2L2* mutant lung cancer cells were sensitive to inhibition by TAK-228 but not by everolimus or rapamycin [118].

*KEAP1-NFE2L2* mutant tumors are dependent on glutaminolysis and inhibition of glutaminase leads to reduced growth and radiation sensitization of *KEAP1-NFE2L2* mutant tumors [119]. In the BeGIN trial (NCT03872427), the glutaminase inhibitor (CB-839) is being investigated in patients with advanced solid tumors with mutations in *KEAP1-NFE2L2*, *STK11*/LKB1, or NF1. In addition, CB-839 is also being tested in a phase II randomized trial (NCT04265534) to compare it with chemotherapy and pembrolizumab as standard-of-care in patients with advanced *KEAP1-NFE2L2* mutant non-squamous NSCLC.

### 4.2. STK11

*STK11* is a key upstream activator of AMP-activated protein kinases that can affect the regulation of glucose and fat *Met*abolism, cell growth and homeostasis [120]. *STK11* mutations occur in 8–39% of NSCLC [121] (Figure 1).

Because both *STK11* and *KEAP1* mutations are highly prevalent in NSCLCs with *KRAS* mutations, *STK11* and *KEAP1* mutations have mainly been evaluated in *KRAS*-mutated patients. Mutations in *STK11* and *KEAP1* genes, as well as the presence of other mutations, are associated with poor outcomes in patients treated with ICI for NSCLC. Skoulidis and colleagues found that as compared to *KRAS* wild-type, concurrent *STK11* mutations result in significantly reduced ORR, PFS, and OS [122].

Similar to *KEAP1*, *STK11* may affect the prognosis of NSCLC. Recently, a retrospective analysis was performed on 2276 patients with NSCLC who received different types of therapy, including *PD-1*/*PD-L1* in beta-blockers, chemotherapy, and tyrosine kinase inhibitors. Regardless of whether *KRAS* mutations were present or absent, *STK11* and *KEAP1* mutations were associated with poor prognoses in all treatment categories. In patients with both *STK11* and *KEAP1* mutations, the prognosis was worse than in those with only one mutation, suggesting an additive effect [123].

However, *STK11* mutations in NSCLC have not been studied clinically, so a comprehensive evaluation of these patients is not be possible.

## 5. Immune Targets

### 5.1. PD-1 and PD-L1

In 1992, researchers identified a novel programmed death-1 (*PD-1*) growth factor in mouse T cell tumors. *PD-1*, which belongs to the regulatory B7-CD28 receptor superfamily, is considered to be a novel kind of cell protein that is widely expressed on the surface of T cells and is closely related to apoptotic growth factors [124]. The main ligand of *PD-1* is programmed death ligand 1 (*PD-L1*), and the two exert immunomodulatory effects through binding. Studies have demonstrated that *PD-L1* expression is elevated in a range of malignancies, including NSCLC, and that this is linked to shorter patient terrible prognosis and survival [125]. In recent years, immune checkpoint inhibitors (ICIs) have exerted antitumor effects through the mechanism of immune checkpoints, especially targeting the *PD-1* and *PD-L1* pathways. However, the selection of the beneficiary population remains a challenge.

*PD-1* is targeted by the monoclonal antibody nivolumab, which is composed of all-human IgG4 antibodies [126]. In several open-label trials (CheckMate 057, CheckMate 026, CheckMate 017) [127,128,129], the results of the trials showed that nivolumab has a longer OS and better safety profile than chemotherapy. A pooled analysis of CheckMate 017 and CheckMate 057 studies showed that five-year pooled OS rates of nivolumab compared to docetaxel were 13.4% vs. 2.6% and 5-year PFS rates of nivolumab compared to docetaxel were 8.0% vs. 0%. Furthermore, nivolumab had a 28% lower death rate than docetaxel, and its adverse reactions were also lower than docetaxel’s.

Pembrolizumab, a *PD-1* receptor agonist, inhibits the interactions between *PD-L1* and *PD-L2*. KEYNOTE-001 showed an ideal result in that the median OS was 22.3 months in treatment-naive patients and 10.5 months in previously treated patients [130]. KEYNOTE-024 then identified pembrolizumab as an effective first-line treatment for patients with advanced NSCLC with *PD-L1* TPS ≥ 50% [131]. Immediately after KEYNOTE-024, KEYNOTE-042 further expanded the population considered to benefit from pembrolizumab treatment [132]. KEYNOTE-042 suggested that pembrolizumab could be extended as first-line therapy to patients with locally advanced or metastatic NSCLC.

Inhibiting *PD-1* and *PD-L1* receptor interaction by targeting *PD-L1* protein, atezolizumab is an IgG1 monoclonal antibody. In an open-label tail called POPLAR which enrolled 287 patients with advanced NSCLC who had progressed after platinum-based chemotherapy. Atezolizumab group had a better medium OS of 12.6 months than docetaxel group whose medium OS was 9.7 months [133]. Subsequently, a multicenter trial called OAK further demonstrated the efficacy and safety of atezolizumab. The median OS of atezolizumab was 13.8 months versus 9.6 months compared to docetaxel [134]. Patients treated with atezolizumab had fewer grade 3 or 4 adverse events compared to patients treated with docetaxel. In 2020, a trial called IMpower110 showed atezolizumab as a first-line treatment with NSCLC with *PD-L1* expression, regardless of histologic type [135].

Avelumab is a monoclonal antibody that targets *PD-L1* in humans. According to its use in first-line treatment of patients with advanced NSCLC, avelumab showed safety and tolerability when expressing antitumor activity [136]. In this trial, the median PFS was 4.0 months (95% CI, 2.7–5.4 months) and the 6-month PFS rate was 38.5% (95% CI, 30.7–46.3%). The median OS was 14.1 months (95% CI, 11.3–16.9 months) and the 12-month OS rate was 56.6% (95% CI, 48.2–64.1%). In patients with platinum-treated *PD-L1*+ NSCLC, avelumab did not significantly prolong OS as compared with docetaxel in the primary analysis [137], a 2-year follow-up data of JAVELIN LUNG 200 showed that OS rates were doubled with avelumab in subgroups with higher *PD-L1* expression (greater than or equal to 50% and greater than or equal to 80%) [138].

Another *PD-L1* inhibitor, durvalumab, targets CD80 and *PD-1* by blocking its binding to *PD-L1*. In a study called PACIFIC of durvalumab after chemoradiotherapy in stage III NSCLC [139]. The median PFS from randomization was 16.8 months (95% CI, 13.0–18.1 months) with durvalumab versus 5.6 months (95% CI, 4.6–7.8) with the placebo (HR, 0.52 [95% CI, 0.42–0.65]; *p* < 0.001). The study reconfirms that durvalumab therapy resulted in significantly longer OS than the placebo [140]. The latest OS and PFS data from PACIFIC further validate the benefit of durvalumab in NSCLC [141,142,143,144].

By inhibiting *PD-1*, cemiplimab stimulates an anticancer response. In EMPOWER-Lung 1, a multicenter trial which examined cemiplimab of advanced NSCLC with *PD-L1* of at least 50%, cemiplimab significantly improved OS and PFS compared with chemotherapy [145]. The median OS of cemiplimab group (*n* = 283) was not reached (95% CI, 17.9 months to not evaluable), while it was 14.2 months in those who received chemotherapy (*n* = 280). Moreover, cemiplimab had a higher median PFS than chemotherapy (8.2 moths vs. 5.7 months). A new treatment option is available for advanced NSCLC with *PD-L1* of at least 50%.

Tislelizumab is a novel *PD-1*-targeted antibody designed to have reduced binding to Fc-gamma receptors, in an effort to overcome a potential mechanism of resistance to *PD-1* inhibitor antibody therapy [146].

Similar to the targeted drugs above, immunotherapy drugs are also inevitably limited by drug resistance. Host, tumor cells, and the immune microenvironment interact to cause tumor immunotherapy resistance [147]. Currently, the most important and effective method of delaying or reversing immune resistance is through a combination therapy strategy [148].

### 5.2. CTLA-4

Another important immune checkpoint is *CTLA-4*. *CTLA-4* is a ligand or receptor for tumor cell-immune cell interactions and acts as an immunomodulator [149]. Treatment against *CTLA-4* plays an important role in melanoma [150]. Similar to *PD-1*/*PD-L1*, ICI anti-CTLA4 has a significant role in NSCLC [151].

Ipilimumab, an inhibitor of *CTLA-4*, may promote T-cell activation and subsequent anti-tumor immunity. In an early study involving patients with advanced NSCLC, median OS with nivolumab plus ipilimumab was 17.1 months (95% CI, 15.0–20.1 months) compared with chemotherapy that was 14.9 months (95% CI, 12.7–16.7 months) [152].

In CheckMate 227, Patients with advanced NSCLC who received first-line nivolumab plus ipilimumab for a minimum of four years have continued to show durable long-term efficacy even after stopping immunotherapy for at least two years. The four-year OS rate with nivolumab plus ipilimumab was 29%, while chemotherapy was 18% [153].

Tremelimumab is a fully human anti-*CTLA-4*, IgG2 mAb. In a phase Ib trial, tremelimumab was initially studied in combination with the anti-*PD-L1* drug durvalumab in the treatment of advanced NSCLC [154].

As a result of these findings, the Phase III MYSTIC Trial was conducted, which tested durvalumab alone or in combination with tremelimumab as an upfront strategy for patients with advanced NSCLC. The median OS was 16.3 months (95% CI, 12.2–20.8 months) in the durvalumab group versus 12.9 months (95% CI, 10.5–15.0 months) in the chemotherapy group (HR, 0.76 [97.54% CI, 0.56–1.02]) [155].

## 6. Combined Immune-Therapies with Targets

### 6.1. STING

Although immunotherapy in combination with chemotherapy is the current standard of care for NSCLC, clinical response varies. The stimulator of interferon genes (*STING*) may be activated by cytosolic DNA fragments [156].

By activating *STING*-mediated antitumor, DNA damage and anti-*PD-L1* synergistic immune responses can be induced by treatment with DNA damage response inhibitors in the syngeneic model of SCLC [157]. Based on this study, Carminia and colleagues explored the profile of immune-related genes related to *STING* pathway activation in NSCLC. The considerable result is that *STING* pathway activation in NSCLC is characteristic of predictive immunotherapy response and is enhanced by cisplatin treatment, suggesting that possible predictive biomarkers and mechanisms can improve response to chemoimmunotherapy combinations [158].

In a study which explores the overall prognosis of NSCLC, the investigators found that high *STING* expression was associated with improved OS in patients with local adenocarcinoma and may be a potential biomarker for NSCLC [159].

### 6.2. LAG3

Lymphocyte-activation gene 3 (*LAG3*) is the most promising immune checkpoint next to *PD-1* and *CTLA-4*. Fibrin-like protein 1 (FGL1), a liver-secreted protein, is a major LAG-3 functional ligand independent of MHC-II [160].

In NSCLC, the high expression of FGL1 and *LAG3* was associated with a poorer five-year OS, respectively [161]. In metastatic NSCLC characterized by high plasma FGL1 levels, the poor outcome after anti-*PD-1* treatment suggests that FGL1 may play a role in tumor immune resistance. Higher FGL1 expression may induce resistance to gefitinib [162].

A novel anti-*LAG3* drug, ieramilimab (*LAG*525), is currently being investigated in two trials as a monotherapy or in combination with PDR001, an experimental anti-*PD-1* drug. Patients with solid tumors, including NSCLC, were tested for safety and antitumor activity of *LAG*525 as a single agent or combined with PDR001. Twelve patients responded to the combination with a durable response and an acceptable safety profile [163].

An ongoing phase II study combining pembrolizumab and IMP321 to treat patients with untreated, unresectable, or metastatic NSCLC is investigating this combination [161].

### 6.3. TIGIT

*TIGIT* (T-cell immunoreceptor with Ig and ITIM domains) gene encodes immunoglobulin proteins of the poliovirus receptor family of inhibitory immune receptors [164].

Overexpression of *TIGIT* in NSCLC may be associated with increased levels of other immunosuppressive receptors (such as *PD-1* and *LAG-3*) and decreased levels of activator receptors [165].

CITYSCAPE is a phase II, randomized, double-blind, placebo-controlled trial. 135 eligible patients were randomly assigned to receive tiragolumab (a *TIGIT* inhibitor) plus atezolizumab (*n* = 67) or a placebo plus atezolizumab (*n* = 68). At a median follow-up of 5.9 months, 11 patients had an objective response in the tiragolumab plus atezolizumab group versus the placebo plus atezolizumab group. The median PFS was 5.4 months in the tiragolumab plus atezolizumab group, and 3.6 months in the placebo plus atezolizumab group [166]. Based on these data, tiragolumab plus atezolizumab is a promising immunotherapy combination for the treatment of previously untreated locally advanced unresectable or metastatic NSCLC.

Studies are ongoing to evaluate the efficacy and safety of pembrolizumab (MK-3475) PLUS chemotherapy in combination with anti-*TIGIT* vibostolimab (MK-7684), MK-5890, or MK-4830 in newly treated participants with advanced SCC or non-SCC (NCT 04165798).

## 7. Molecular Testing

With so many authorized targeted drugs, molecular profiling must be quick and accurate. Despite this, genetic testing is still not widely used in clinical practice.

Single biomarker assays have traditionally been performed using immunohistochemistry, fluorescence in situ hybridization, polymerase chain reaction, and Sanger sequencing methods [167]. Since it is relatively easy to implement, has a quick turnaround time, and is relatively inexpensive, it is suitable for a wide range of applications. However, as the number of molecular biomarkers grows, multiplexed assays, particularly NGS, are becoming more popular for evaluating several biomarkers in a single workflow. Tissue yield and attrition are important factors in NSCLC, as diagnostic specimens are frequently acquired from tiny biopsies [168]. As a result, the use of liquid biopsies to analyze circulating tumor DNA has exploded. However, emerging assays must be compared to existing molecular tests to demonstrate their clinical validity, practicability, and cost-effectiveness. In recent years, several studies have revealed that NGS is cost-effective in treating LUAD when biomarkers other than *ALK*, *EGFR*, and *ROS1* are analyzed [169]. Even in Asian cultures, where *EGFR* mutations are the most common driver mutation, this has been confirmed [170]. However, before implementing this in various settings, health-care resource use, prescription prices, and long-term patient outcomes must be considered. NGS has already been found to have superior sensitivity when compared to single-gene focused assays [171,172,173]. Current molecular tests for oncogenic drivers are shown in Figure 3.

It is possible that many new biomarkers have no recognized gold standard methods of detection. Furthermore, there may be discrepancies between tests, such as for *RET* rearrangements, RNA-based vs. DNA-based NGS for *MET* exon 14 alterations [174] and NGS vs. fluorescence in situ hybridization [175]. Tests based on NGS panels, such as those in tissue and plasma, have been approved as companion testing for several targeted medicines.

## 8. Discussion

Despite the current good progress in the treatment of NSCLC at the time of writing, FDA-approved therapeutics are *EGFR*, *ALK*, *ROS1*, *BRAF^V600E^*, *MET*, *RET*, *NTRK*, *KRAS*^G12C^, *HER2* and *PD-1* molecular abnormalities [176] (Figure 2). NSCLC continues to have the highest incidence and mortality rate of any disease [1].

When compared to chemotherapy, biomarker-targeted therapies have demonstrated great potential in NSCLC, drug resistance is a problem that never goes away [177,178]. As a result, a deeper understanding of the molecular basis of drug resistance is critical in order to delay or even prevent resistance.

Despite these challenges, the future aspect of the NSCLC treatment landscape does look bright, with multiple promising new therapies targeting newly labeled proteins and less prone to acquiring resistance, many of which are even in clinical studies. Innovative therapies targeting other immune checkpoint receptors, such as Toll-like receptors 9 (DV281 and CpG ODN) [179,180], various neoadjuvant therapies, combined immune-therapies have shown significant results in preclinical studies and are currently under clinical investigation.

With a better understanding of the molecular heterogeneity of NSCLC and an increased clinical use of NGS, the development of novel targeted drugs will continue to accelerate. Recent deep multiset sequencing of oncogenic drivers, such as proteome sequencing, may reveal further molecular signatures related to response and resistance to therapies and novel targets [181]. These features will allow the development of therapies with higher selectivity and specificity for targets of interest.

In summary, the identification of actionable targets in NSCLC has changed the way many patients are managed and treated. A growing number of active phase II/III studies are showing promise in combating medication resistance and discovering new viable drugs. The current research into the genetic and epigenomic basis of cancer ecosystems will lead to the development of new, highly specific targeted drugs. In a word, biomarker-target therapies are potential and powerful weapon to treat even overcome NSCLC.

## Figures and Tables

**Figure 1 cells-11-03200-f001:**
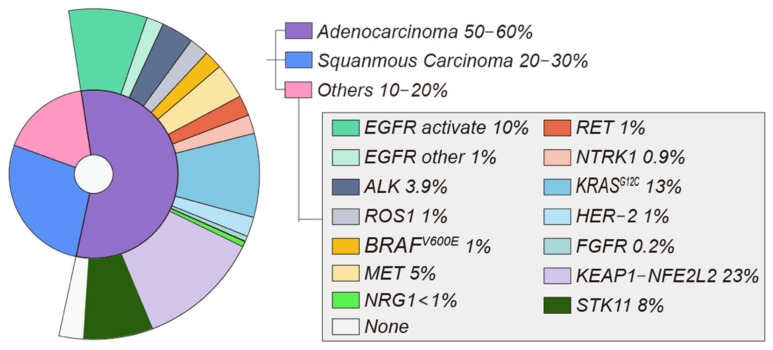
In total, 50–60% of NSCLC cases are adenocarcinomas, whereas 20–30% are squamous cell carcinomas. Adenocarcinoma molecular alterations related to targetable oncogenic drivers. Incidences of oncogenic driver alterations extracted from study results [6,9,10].

**Figure 2 cells-11-03200-f002:**
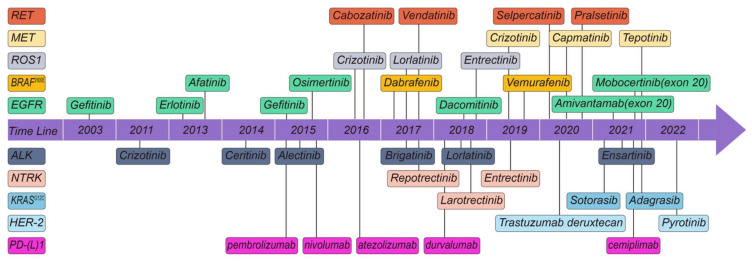
Timeline of targeted drugs which FDA approved for oncogene-driven NSCLC.

**Figure 3 cells-11-03200-f003:**
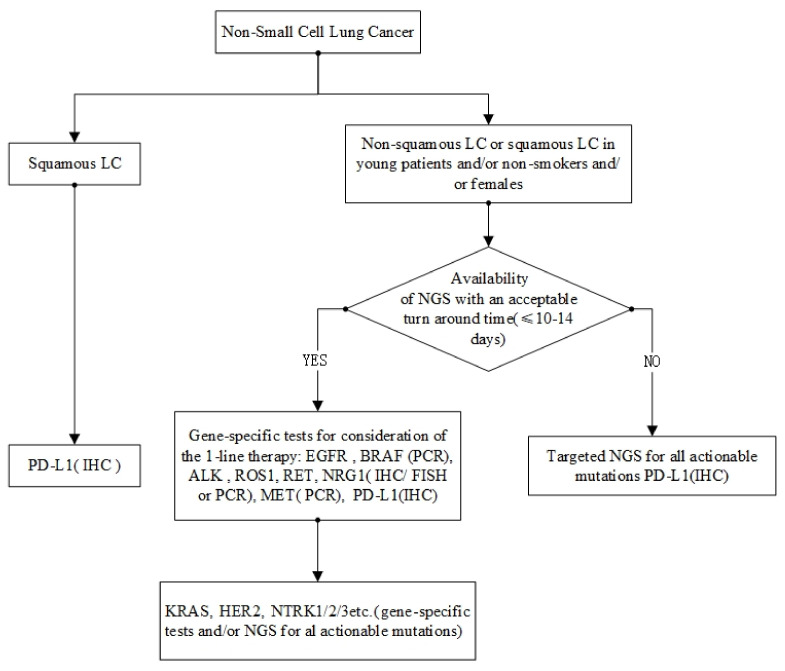
Current molecular tests for oncogenic drivers.

**Table 1 cells-11-03200-t001:** Part of the key randomized trials in *EGFR*-positive NSCLC.

Drugs	Trails	Comparator	ORR, %	PFS, HR	OS, HR	Ref.
Gefitinib	WJTOG-3405	Cisplatin plus docetaxel	62.1 vs. 32.2	0.489	1.252	[13]
Erlotinib	OPTIMAL	Carboplatin plus gemcitabine	83 vs. 36	0.16	1.19	[14]
Afatinib	LUX-Lung-7	Gefitinib	72.5 vs. 56	0.73	0.86	[16]
Osimertinib	FLAURA	Erlotinib or gefitinib	80 vs. 76	0.46	0.80	[15]
Dacomitinib	ARCHER-1050	Gefitinib	75 vs. 72	0.59	0.760	[17]

Abbreviations: ORR, objective response rate; PFS, progression-free survival; OS, overall survival.

**Table 2 cells-11-03200-t002:** Part of the key randomized trials in *ALK*-positive NSCLC.

Drugs	Trails	Comparator	ORR, %	PFS, HR	OS, HR	Ref.
Crizotinib	PROFILE-1014	Platinum plus pemetrexed	74 vs. 45	0.45	0.76	[42]
Ceritinib	ASCEND-4	Platinum plus pemetrexed	72.5 vs. 26.7	0.55	0.73	[44]
Alectinib	ALEX	Crizotinib	82.9 vs. 75.5	0.5	0.67	[45]
Brigatinib	ALTA-1L	Crizotinib	71 vs. 60	0.49	0.92	[46]
Lorlatinib	CROWN	Crizotinib	76 vs. 58	0.28	0.72	[47]
Ensartinib	eXalt3	Crizotinib	75 vs. 67	0.52	0.88	[48]

Abbreviations: ORR, objective response rate; PFS, progression-free survival; OS, overall survival.

## Data Availability

Not applicable.

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
