# Peer review of "Biomarker-Targeted Therapies in Non–Small Cell Lung Cancer: Current Status and Perspectives"

_cells, 2022, doi:10.3390/cells11203200_

Round 1
Reviewer 1 Report (Previous Reviewer 1)
I think, all specific comments from the previous submissions have been addressed appropriately. The manuscript is now up-to-date.
Reviewer 2 Report (Previous Reviewer 3)
The authors described the current knowledge on molecular changes in NSCLC that are targets for therapies, and future developments in biomarker therapies in precision medicine. The manuscript “Biomarker-targeted Therapies in Non-small Cell Lung Cancer: Current Status and Perspectives” is well-written and easy to follow.
Overall, the content of the manuscript was comprehensive. The revised manuscript is much better than before. I believe this manuscript has been significantly improved and now warrant to publish in cells.
This manuscript is a resubmission of an earlier submission. The following is a list of the peer review reports and author responses from that submission.
Round 1
Reviewer 1 Report
This is a review of targets and therapies in Non-Small Cell Lung Cancer. As it stands the review does not add much to many other reviews that have been published previously and is actually quite otdated. Thus, I believe it may only be published (if at all) after extensive revisions.
General remarks:
1. The abstract is completely meaningless and could have been published 10 years ago. Thus, I suggest to refer to a more specific topic, i.e. review the multitude and sequence of targeted inhibitors for oncogeneic drivers in comparison and review data on what therapy first, second third-line depending on resistance mechanisms.
2. All therapies for advanced-stages NSCLC are not curative and eventually elicit resistance with emerging tumor progression. Thus, all chapters on biomarkers (EGFR, ALK, ROS etc..) need to address the challenge of determining resistance mechanisms and delivering further effective targeted therapies (also true for immun-therapies). This important area is almost completely blined out in the manuscript
3. A huge area of innovation and improvement is neo-adjuvant therapies and targeted adjuvant therapies. This is not mentioned at all in this review and needs to be covered.
4. Another area essentially not covered is metabolic targets (KEAP1, STK11 mutated tumors) and combined immune-therapies with targetes (STING, LAG3 etc). This needs to be added, if the title "Current status and Perspective" should apply.
Specific comments:
p1, line 42ff: "At present, the actionable mutations with approved therapies include:..." There are obviously significant differences in approval between different countries world-wide, however, the authors later refer to US FDA approvals and hence this list (if it applies to FDA) is not complete. Missing here is: NRG1, KRAS, emerging is KEAP1, MAPK1.
p3, line 114ff. The issue of rare, unusual EGFR mutations is quite important and not well covered here. Recently two large studies from the German nNGM network and from the US have been published on classification and actionability of rare EGFR mutations in Annals of Oncology and JCO. This data should be carefully reviewed and included and go far beyond the quite small and old studies mentioned by the authors.
p5, line 186ff: It is well established that the specific types of ALK fusions (long versus short transcripts and co-occurring p53 mutations provide significant risk factors for non-actionable resistance mechanisms. This data must be reviewed here as they modifiy treatment options.
p6, line 231ff: It is well established that Crizotinib, Entrectinib and Lorlatinib are good options for patients with ROS1 fusions. Here, the review should cover the subject of how to select patients for the best sequence of these options (primary brain mets vs primary extracranial mets, selection of side-effects where Crizo is much better tolerated that Lorlatinib during long-term treatment etc).
p7,line292ff: Here, the issue of MET-resistance is completely missing and the option of selecting type-I versus type-II inhibitors. Also, the issue of treatment of high-level MET amplification and MET-fusions is missing. Data are available from clinical trials (Geometry and others) and should be reviewed properly.
p10: Data from AKIP1 and MXRA5 should be deleted, unless clinical trial data are available for review (not known to me). Otherwise a huge number of potential targets, currently not actionable, must be added as well as PIK3CA and other mutations.
Here, I strongly suggest to add KEAP1 mutations, driving NRF2 activation, as clinical trial data are available using glutaminase inhibitors and others.
The chapter on mono-immune therapies and chemo-immune therapies are quite outdated. Completedly missing are trial data from combined checkpoint inhibitors (STING, LAG3, and others) and also important data from neo-adjuvant immune therapy trials.
Reviewer 2 Report
There are several points that need to be revised:
- Please, you should revise the English language of manuscript and improve the form and the English writing explanation of the concepts.
-In the section Abstract, page 1-2 at line 23 and at line 48, you talk about your work as an “editorial” but it is submitted as a “review”, please verify.
-Please, you should increase the references in the whole manuscript, especially to verify the trials, ratios and percentages you mention or should mention; for example there should be some reference for the period “Approximately 60% of patients with advanced NSCLC subtypes have specific molecular changes that may be suitable for target therapy” in page 1 line 36-38 that justifies that percentage.
- The nomenclature of the mutations should be reported following HGVS nomenclature (EGFR p.L858R) in the whole manuscript. In addiction the genes should be reported italics.
- Please, you should also include in the introduction section and not only in section 4 "Immune Target" the PD-L1 target and its therapy since it is a leading marker in the treatment of non-small cell lung cancer.
- In the section “Molecular Testing”: Since your focus is the biomarker individuation and the appropriate targeted drug, you should discuss more extensively and accurately about the methods that can be used to the biomarker detection, highlighting the pros and cons of each method cited.
- There are some typos that should be revised (i.e. page figure 1 “squanmous” and “carinoma”; page 4 line 160 “brgatinib”; page 5 at line 215 “cricotinib”; page 7 line 315 and 321 “%age”; page 10 at line 420 “NSLCC”);
- In the section 2.7 “NTRK1” you mention both EMA and FDA (page 8 line 352-356): this differentiation should be applied to the whole manuscript.
- In the section 2.8 “KRAS” you should mention the frequency of the mutation in NSCLC like you did for other biomarkers.
- The section 6 Discussion should be revised and refocused. References should be used.
Reviewer 3 Report
The authors described the current knowledge on molecular changes in NSCLC that are targets for therapies, and future developments in biomarker therapies in precision medicine. The manuscript “Biomarker-targeted Therapies in Non-small Cell Lung Cancer: Current Status and Perspectives” is well-written and easy to follow.
Overall, the content of the manuscript was comprehensive. However, I believe that there are important points lacking in this manuscript.
Comments:
Line 42: The authors listed the actionable mutations with approved therapies. Please add the KRASG12C mutation to this list.
Line 66: The two most frequent mutations, “exon 21 mutation” and “deletion of exon 19”, should be described definitively as “L858R” and “a 15 bp deletion in exon 19”, respectively.
Line 130: Lorlatinib should be added to this list of ALK inhibitors.
In section 2.2 ALK: The authors should add the crizotinib-resistance mutations of L1196M and G1269A. This is important because these mutations can lead to the development of a newer generation ALK inhibitors.
In section 2.4 BRAF: Please clearly indicate that the combination of dabrafenib and trametinib is the first FDA-approved treatment specifically for patients with BRAF V600E mutation-positive metastatic NSCLC.
Line 366: Totorasib should be revised to sotorasib.
In Figure 2: The first approval of gefitinib is in 2003.
In section 3.1 HER2: The information of T-Dxd (DS-8201) in HER2-mutant or amplified NSCLC should be included.